# A Bluetooth-Low-Energy-Based Detection and Warning System for Vulnerable Road Users in the Blind Spot of Vehicles

**DOI:** 10.3390/s20092727

**Published:** 2020-05-11

**Authors:** Nick De Raeve, Matthias de Schepper, Jo Verhaevert, Patrick Van Torre, Hendrik Rogier

**Affiliations:** IDLab, Department of Information Technology, Ghent University-imec, Technologiepark-Zwijnaarde 126, 9052 Gent, Belgium; nick.deraeve@ugent.be (N.D.R.); matthias.deschepper@ugent.be (M.d.S.); patrick.vantorre@ugent.be (P.V.T.); hendrik.rogier@ugent.be (H.R.)

**Keywords:** blind spot detection and warning, BLE, bluetooth low energy

## Abstract

Blind spot road accidents are a frequently occurring problem. Every year, several deaths are caused by this phenomenon, even though a lot of money is invested in raising awareness and in the development of prevention systems. In this paper, a blind spot detection and warning system is proposed, relying on Received Signal Strength Indicator (RSSI) measurements and Bluetooth Low Energy (BLE) wireless communication. The received RSSI samples are threshold-filtered, after which a weighted average is computed with a sliding window filter. The technique is validated by simulations and measurements. Finally, the strength of the proposed system is demonstrated with real-life measurements.

## 1. Introduction and Related Work

On Belgian roads, every year approximately 50 people are involved in blind spot accidents [1], of which approximately 10% end lethally. Annually, the government invests a lot to raise the awareness of this problem. However, the danger still exists, mainly due to the lack of reliable communication between the truck driver and the vulnerable road user. Therefore, in this paper we propose a blind spot detection and warning system that makes both parties aware of a potential blind spot danger.

Most systems which are developed and sold on the market are camera-based or radar-based. In most cases, camera systems use visual parameters to detect vehicles in the blind spot through post-processing machine learning algorithms [2,3,4]. Their big advantage is the visualization of a potential accident. At night, when compared to daytime, special camera systems are needed and the detection rate is inevitably reduced. In [5,6] an improved solution for detection at nighttime was introduced. However, since the detection is based on images, all cameras have to stay clean, which is often problematic in the truck’s operating conditions. Furthermore, when a truck makes a turn, the cameras lose the observation position as well as the detection region. In contrast, radar-based systems can be applied in real-life situations [7]. Their biggest advantage is their versatility, since these systems can operate during day- and nighttime. Most studies, however, show that only motorized vehicles are detected, and not vulnerable road users. It is easy to understand that radar detection for vulnerable road users is problematic, due to their fairly small radar cross section in combination with significant clutter on a realistic radar image. In most situations, rain, snow, trash bins, etc., are also detected, leading to many unwanted false positives. In this paper, a BLE-based detection and warning system where both the truck driver and the vulnerable road user are warned for a potential danger, is proposed. Moreover, as the system is based on RF communication, the major problems of camera- and radar-based detection systems are solved.

Publications based on reconstruction reports of heavy good vehicle accidents confirm that most of the vulnerable road users are on the right side of the truck at the moment of the accident [8,9]. In Figure 1 different zones around the truck are visualized. The areas with the double blue lines are visible from the truck driver’s seat, directly through the windows. The areas with orange solid and dashed lines are visible via the truck’s mirrors, when deployed as regulated by law. The areas with the red squares are not visible from inside the cabin through the windows and/or the mirrors. These areas are called the blind spots. Important to mention here is that all areas are drawn based on the position of the truck in Figure 1. When the truck turns to the right, the area visible through the mirrors decreases and the blind spot area increases. Furthermore, mirrors are a passive system and will not alert the truck driver of potential danger. In this paper, we propose a blind spot detection and warning system based on Bluetooth Low Energy (BLE), warning both the truck driver and the vulnerable road user for a possible blind spot accident. Although the system can be used by all kinds of cyclists, the rest of the paper focuses on pedestrians as vulnerable road users.

This paper presents the hardware implementation of the different nodes and the design of a small sensor or wearable. Moreover in Section 2, the communication protocol between the different nodes and the applied filtering are discussed. Next, the performed simulations and real-life measurements that validate the system’s operation are detailed in Section 3 and Section 4. The paper ends with a conclusion formulated in Section 5.

## 2. Design

The system proposed in this paper creates a complete detection area around the truck. Therefore, Figure 1 also shows detection nodes attached to the truck in order to detect objects and persons in the blind spots. Therefore, at the front of the truck a detection node is deployed and also one at the rear of the truck, as well as uniformly distributing three nodes along the right side. The vulnerable road user is equipped with a small sensor or wearable, wirelessly connected to all detection nodes. Hence, the vulnerable road user can be detected when entering the blind spot. In what follows, the detailed hardware implementation of the designed nodes is described and the software routines and filtering algorithms are explained.

### 2.1. Hardware Implementation

The communication between the nodes in the proposed system is based on Bluetooth Low Energy (BLE). This communication standard was selected for the low power capabilities, minimal complexity, low price, easy connectivity with smartphone applications and compatibility of future versions. Furthermore, Silicon Labs provides multiprotocol chips, including BLE, making their hardware suitable for the design of a proof-of-concept.

For this system, BLE4.2 is the minimal required version. All previous versions used a one-to-one communication protocol, while starting from BLE4.2 one-to-many is possible. The entire required data throughput is limited and higher versions of the BLE-standard are not necessary, limiting also the power usage [10].

The design of the hardware relies on BGM111 modules from Silicon Laboratories [11]. These modules use the BLE4.2 stack [12] and contain an on-board 32-bit, 38.4
MHz ARM Cortex M4 [13] microcontroller with DSP instruction set, combined with an integrated antenna. In order to have an optimal maximal range, the data sheet of the BGM111 prescribes an empty space of at least 16 mm around both sides of the System on Chip (SoC). Furthermore, a TAG-connect [14] connector was used to program all different nodes via the SWD protocol [15]. As power supply LiPo batteries of 3.3 V [16] are used.

Figure 2 shows the designed PCB for the detection node. Some LEDs were added for initialization and debugging purposes. As protection PCB plastic boxes were 3D-printed. A neodymium magnet is glued at the bottom side of the box to attach the node alongside the truck.

The wireless starter kit accompanying the BGM111 module from Silicon Laboratories [17] is used as central node. This development board contains a small LCD screen, LEDs, push buttons, etc., but also the necessary circuitry to debug and log all necessary data.

Figure 3 shows the designed wearable and buzzer PCB next to its flexible housing. A light band worn by runners, cyclists and pedestrians, was chosen. The main PCB contains the BLE module and the peripheral components to activate the module and to set the outputs. Also here, the size of the PCB is very important. To fit the PCB inside the package, the width at both lateral sides of the SoC was reduced to 11 mm, yielding a decrease by ±10% in maximum range, according to the data sheet.

The second PCB of Figure 3 contains the buzzer. On this PCB, an oscillator was implemented based on NAND gates with a built-in Schmitt trigger. According to the data sheet of the buzzer [18], a signal between 4 and 8 kHz is necessary to receive the highest pitch. Realized with in-house components, a frequency of 5.4
kHz was measured with a Rigol DS1054Z oscilloscope. Furthermore, LEDs and a vibration motor were added for flashing and vibrating in case of an alert.

### 2.2. Software

In this section, a global overview of the communication steps is given. The sequence diagram in Figure 4 pictures these steps. After initialization, the detection nodes send advertisement packets. The wearable records the RSSI levels of the received packets of the different detection nodes in a database. When the buffer is full, the wearable algorithm calculates the alert level for each detection node. If there is an alert, the wearable makes a connection with the corresponding detection node and alerts it. Afterwards, this detection node immediately disconnects from the wearable and sends a message to the central node to alert it.

At this point, the truck driver and the vulnerable road user are both alerted of each other’s presence. This subsequently enables both parties to take the required measures to avoid a blind spot accident. In the following subsections, the communication steps of the different nodes and the filtering technique implemented in the algorithm are discussed in more detail. Also, a more detailed interpretation of the buffer size and the weights for the alert calculation is given.

#### 2.2.1. Wearable

The designed wearable contains a push button to switch on the system. After the initialization of the complete database, the device starts scanning for advertising messages from the detection nodes [19]. The wearable runs three main software routines. The first one is the initialization routine, the second adds the RSSI levels to the database and a last one checks the database. Figure 5 illustrates in the top part the initialization of the wearable, while the bottom part shows the function to add the RSSI levels to the database. In the first software routine, the LEDs of the wearable light up, while it starts scanning for advertisement messages with predefined data. The second software routine adds the received RSSI levels to the corresponding buffers inside the database. At the same time, a timer-controlled routine runs in the background and checks this database. If no more packages from a specific detection node are received, this timer-controlled routine clears the database entity so it can be reused.

The third and final software routine checks the database, as can be seen in Figure 6. The wearable continuously scans for advertisement packets. In order to have a fast response, the system pauses scanning after every five received packages and then checks the database. In this way, with an advertisement interval of 20 ms, every 100 ms the database with all received RSSI samples in multiple buffers is checked. When one of these buffers is full, the algorithm filters all RSSI samples, calculates the average value and determines the alert level. If the level is ‘high’, the routine connects to the detection node. Once the connection is made, the detection node immediately disconnects from the wearable and sends the corresponding alert to the alert level characteristic of the central node [12]. Because the connection is immediately disconnected, the wearable sets the alert and restarts scanning for new advertisement messages. When an alert-level-characteristic message is received, the central node also sets the corresponding alert.

#### 2.2.2. Detection and Central Node

The next sequence diagram (Figure 7) shows the initialization procedure of the central and detection nodes. When the button on the central node is pushed, the initialization is started and the central node advertises. In a second step, the button on the first detection node is pushed. This detection node starts scanning for the advertising messages from the central node. When this message is received, the detection node makes a connection and requests a handle for the immediate alert service and for the alert level characteristic [12]. These handles contain the addresses of the memory allocations inside the BLE stack. Afterwards, the detection node sends an acknowledgment and starts scanning for ‘start’ packages. The first detection node is now completely initialized and is waiting to start advertising. The same steps are repeated consecutively for every detection node.

When all detection nodes are connected, the central node then advertises ‘start’ and goes into a waiting state. In total, ten ‘start’ messages are used, in order to have redundancy. When receiving these ‘start’ messages, the detection node scans for packages sent by the wearable. During initialization, the central node and detection node save the addresses from the other nodes. The next time the system is activated, the nodes can just connect and start advertising without the need of the complete initialization process. If a node fails, the addresses can be adapted by reinitializing the complete system.

### 2.3. Filtering

The detection system has to rely on very volatile RSSI levels, hence requiring extensive filtering. The recorded signal levels are influenced by several radio-wave-propagation effects. Path loss causes a gradual attenuation of the signals when the distance between the transmit and receive antennas increases. Shadowing and multipath fading cause these signals to fluctuate significantly during the recorded trajectory. Shadowing is inevitably caused by the human body on which the wearable is worn. As the body is situated into the radio-wave-propagation path, signals are variably attenuated depending on body orientation and posture. While walking, the arm on which the wearable is worn also moves, causing signal fluctuations. Additionally, multipath fading results in quick signal variations. The physical reason for this phenomenon is the interference between signals that travel along different paths. Metal obstacles cause the strongest reflections, but also trees and buildings play an important role. The operating frequency of the detection system is 2.45
GHz, corresponding to a wavelength of 12 cm. Due to alternating destructive and constructive interference the signal fluctuates rapidly, even with small displacements of the transmitter and/or receiver.

A lot of research was performed in order to improve the detection based on RSSI levels and to mitigate radio propagation related effects [20,21]. Most of these algorithms require a lot of resources in terms of calculations and memory, hence they are mostly performed in post-processing. To retain real-time behavior, two basic filtering systems are proposed. The first filter was implemented in order to suppress outlier RSSI samples due to path loss, shadowing and fading. In this filter received RSSI samples smaller than a certain threshold level are replaced by that threshold value.

The second filter is a weighted average filter with sliding window [22]. This filter is implemented in order to smoothen the received RSSI samples and to calculate the average value, as is shown in Formula (Equation 1). All values in these buffers are expressed in dBm.
(1)$$RSSI_{avg}=w_{1}.\frac{\sum_{i=Q_{2,3}}RSSI_{sort,i}}{\left\lfloor \frac{k}{2}\right\rfloor }+w_{2}.RSSI_{k+1}$$

The proposed filter consists of a buffer that is filled with RSSI samples. When the buffer is completely full, the buffer is sorted ($RSSI_{sort}$) and the average is calculated with the RSSI samples in the interquartile range between the 0.25 and 0.75 quartile, visualized by $Q_{2}$ and $Q_{3}$ in Figure 8. The size of the buffer is represented by *k* and the summation of the selected RSSI samples is divided by the floored value of k/2. Afterwards, the average is multiplied by the weight $w_{1}$. In a next step, a new RSSI sample ($RSSI_{k+1}$) is multiplied by the weight $w_{2}$ and added to the average of the sorted buffer. These weights are calculated based on Formulas (Equation 2) and (Equation 3). Figure 9 illustrates how the weights are generated. $x_{1}$ and $x_{2}$ represents the number of RSSI samples that are being used to calculate the weight. The denominator is half the size of the buffer plus one, because of the last added RSSI sample ($RSSI_{k+1}$).
(2)$$w_{1}=\frac{x_{1}}{\left\lfloor \frac{k}{2}\right\rfloor +1}$$
(3)$$w_{2}=\frac{x_{2}}{\left\lfloor \frac{k}{2}\right\rfloor +1}$$

## 3. Measurements

In this section, two measurement setups are analyzed and different simulations are explained in detail. From these simulations, the parameters for the final algorithm were extracted. To conclude this section, results with the optimized algorithm are shown.

### 3.1. Measurement Setup

#### 3.1.1. Static RSSI Measurement

A first measurement campaign was set up to find the most appropriate threshold level. It is important to note that the accurate conversion from RSSI levels to distances is not possible, but ranges of RSSI levels corresponding to different distances can be determined. To obtain this feeling, some static measurements were performed and are schematically illustrated in Figure 10. A detection node was fixed against a metal container building at a height of 1.2
m (as is shown in Figure 11). This metal structure replicated a metal trailer of a truck. At a distance of 1 m, 250 RSSI samples were logged while the test person was standing still. The same measurement was repeated at distances varying from 1 m to 8 m in steps of 1 m.

Next, the measured data is filtered with a moving average filter. A window size of 7 measurements was chosen because, given the measurement rate, the average speed of the user and the small-scale fading pattern occurring at 2.45 GHz, this window size offered the best compromise between sufficient smoothing and limited delay. Figure 12 shows the filtered data at each distance, together with the unfiltered data as dots. There is a clear difference in RSSI levels between 1 m and 2 m. Starting from 3 m, overlapping ranges occur. Since the system has to start detecting vulnerable road users at a minimum distance of 8 m or more, a threshold level of −65dBm was selected. At a distance of 3 to 4 m, the system has to give an alert. So an RSSI alert level of 10 dBm higher was selected.

#### 3.1.2. Dynamic RSSI Measurement

To obtain the optimal parameters for Formulas (Equation 1)–(Equation 3), a number of dynamic RSSI measurements are performed. Figure 13 shows the measurement setup. Also here, the detection node was deployed on a metal container building at a height of 1.2
m. To simulate a vulnerable road user walking beside a truck, a test person walks at a distance of 4 m in front of the metal container building, starts at 6 m before and ends 6 m beyond the node, covering a trajectory of 12 m. In three different runs, RSSI samples, received by the test person, are logged and used in the following simulations to determine the parameters of both filters.

### 3.2. Threshold Filtering

The first filter compares each RSSI level to a fixed threshold value, as is defined in Section 3.1.1. The comparison itself can be done via different filtering techniques. The first technique is the minimum threshold filtering: every RSSI value lower than the threshold value is replaced by the threshold value. The second proposed technique is the step 1 dB filter: every RSSI value lower than the threshold is replaced by the previous value minus 1 dB. The same idea is used for the third technique with a step of 2 dB, although the RSSI measurement resolution of the used hardware is 1 dB.

Figure 14 presents the RSSI samples filtered by the three techniques, the original data and the discarded data points. The optimal threshold has already been derived earlier, and is represented as a solid black line. Starting from sample 55, the effect of the different techniques becomes visual. The step 1 dB (dashed green line with right triangles) and 2 dB (dashed blue line with left triangles) techniques result in RSSI levels with a much lower value than the original ones. Especially in areas with many discarded points, the negative influence becomes even larger. In contrast, the minimum filtering technique yields the most acceptable result. All discarded points are replaced by the threshold value which results in a more acceptable effect. Therefore, the minimum technique was selected as the preferred filtering technique for the first filter.

### 3.3. Weighted Average Filter with Sliding Window

For the weighted average filtering given by Formula (Equation 1), a number of simulations were performed to find the best suited buffer size and weights. In order to find these parameters, the best result of the buffer size is used in the simulations for the weights and vice versa. Figure 15 shows the minimum threshold filtered data that is used in every next simulation.

In Figure 16, it can be seen that the size of the buffer has a large influence indeed. This size needs to be chosen carefully: the larger the buffer, the more smoothing effect. However, the buffer has to be filled in a reasonable time. Advertisement packets are sent at 20 ms intervals. For a buffer size equal to 51, it takes 1.02
s to calculate the first alert. For small buffer sizes, the obtained average value is by far too small compared to the original data points of Figure 14. Therefore, a buffer size of 31 was selected as the best compromise, offering an acceptable delay in combination with sufficient sensitivity. In case of an alert, a connection is made to the corresponding detection node, alerting it. This alert is forwarded by the detection node to the central node, resulting in a connection time or latency limited to 620 ms.

For the weight, a similar conclusion can be drawn. Figure 17 demonstrates the effects of different weights. Since the buffer contains the most information, 50% as a lower bound for $w_{2}$ was set. Earlier a buffer size of 31 had been selected, resulting in an average calculation based on the middle 15 values (between the 0.25 and 0.75 quartile). The denominator of Formulas (Equation 2) and (Equation 3) was set to 16. As can been seen in the figure, 68.75% or 11/16 for $w_{1}$ gives the smoothest result. Furthermore, this value still attributes sufficient weight ($w_{2}$) to the most recently added sample (31.25% or 5/16).

### 3.4. Optimized Algorithm

In order to verify the selected threshold filtering, buffer size and weights, the simulations were repeated for three other data sets. All runs were filtered separately and the average value is presented in the graphs, where the measurement spread with minimum and maximum for the selected parameters is also represented.

The result of the selected threshold filter is displayed in Figure 18. For every RSSI sample the minimum and maximum value of all runs are displayed, showing the performance of this threshold filter.

Figure 19 shows the variation in RSSI levels for a buffer size of 31. Taking the measurement spread into account, similar results are obtained.

Figure 20 demonstrates the result of the weight simulation for the three runs. The measurement spread for 68.75% has an acceptable effect for the different data sets.

## 4. Real-Life Measurements

Based on real-life measurements, the performance of designed hardware and the realization of the optimized algorithm are validated. After the description of a general test, also the system behavior for a big group of people is handled.

### 4.1. General Test

As a general test, the system was mounted on a truck and multiple secondary school teenagers were equipped with a wearable, worn on the left upper arm. One at a time, a pupil approached the truck from the back or the front. The start position of everyone was 20 m away from the truck. From the moment the pupil started to walk towards the truck, the wearable was activated. Figure 21a shows a pupil receiving his second alert, with the wearable blinking.

During this test, the general operation of the system was validated. At the start of the test, it was experienced that −65dBm is a little bit too low. Therefore, the threshold level was changed to −70dBm. This increased the detection distance by ±3 m and when starting from the rear end of the truck, the first alert was received at a distance of around ±8 m. Coming from the other side, the alert was received much later, resulting in a distance of ±3.5 m from the truck. This can be explained by the fact that the wearable was worn at the left upper arm and was hence oriented away from the detection nodes.

### 4.2. Large Group of People

Other tests were carried out with more than one person at a time: with five, ten and twenty pupils respectively.

Five persons walk in a small group towards the truck starting from ±20 m from behind the truck (Figure 22). The first alert was received by persons in position one and two, followed by an alert for the pupil in the third position. The pupils in position four and five received the alert from the moment they passed the others. This test shows that the RSSI levels for pupil four and five are influenced by the others around them. When there is a Line-of-Sight (LoS) connection, the system performs perfectly.

The test with a group of ten was in a random position. As can been seen in Figure 21b, the first time the teenagers walked towards the truck starting from 20 m behind the truck and the second time from the front of the truck. Pupils with an LoS connection received the earliest alert just-in-time. Repeated with twenty persons (see Figure 23a), alerts were received at various times. Once the teenagers started before the truck (see Figure 23b) and the wearable on the left upper arm was obstructed by the different bodies, alerts were received much later. However, all people on the side of the truck received a fast alert.

### 4.3. Verification Measurement

In order to test the system progressively, the same tests were repeated with a larger truck (±16 m) and 30 other persons (see Figure 24a,b). This way, the entire system was tested with a larger group and employing the longest truck allowed on Belgian roads. Just like in previously described real-life measurements, the test persons walked by in different group sizes and positions. Also here, the proposed system performed as intended. For students starting from the back, the first alert was received at a distance of approximately 8 m, measured from the back of the trailer and confirming the results described above.

## 5. Conclusions

In this paper, a blind spot detecting and warning system based on BLE wireless communication is proposed, relying on RSSI measurements. The system consists of five detection nodes around the truck which advertise their presence. The vulnerable road user uses a wearable that scans these advertisement packets. The algorithm inside the wearable interprets these messages and applies filtering on their RSSI levels.

The algorithm itself consists of two filters: the threshold filter and the weighted average filter with sliding window. Based on static RSSI measurements, the threshold level was fixed at −65dBm. Later, during the real-life measurements, this value is lowered to −70dBm. From the threshold simulations, the minimum technique is selected as the preferred threshold filter. Dynamic RSSI measurements are performed to find the best suited buffer size and weights to be used. A buffer size of 31 is proposed and for the weights $w_{1}$ and $w_{2}$ the values 68.75% or 11/16 and 31.25% or 5/16 are suggested, respectively. The first alert is received in 0.62
ms.

During the real-life measurement, the system performed reliably well. The first alert for a vulnerable road user starting from the back of the truck is received at ±8 m. The test with multiple vulnerable road users at the same time lead to the same results. When the wearable is surrounded by many people, the alert is received slightly later. In a group of people, only a few need to wear the wearable in order to receive an alert, the complete group will be alerted due to the light and sound effect of the others.

A blind spot detection and warning system is proposed, relying on Received Signal Strength Indicator (RSSI) measurements and Bluetooth Low Energy (BLE) wireless communication. Compared to camera- and radar-based systems, the proposed system is based on RF communication and uniquely identifies all vulnerable road users. The system warns both the truck driver and the vulnerable road user about a potential danger.

## Figures and Tables

**Figure 1 sensors-20-02727-f001:**
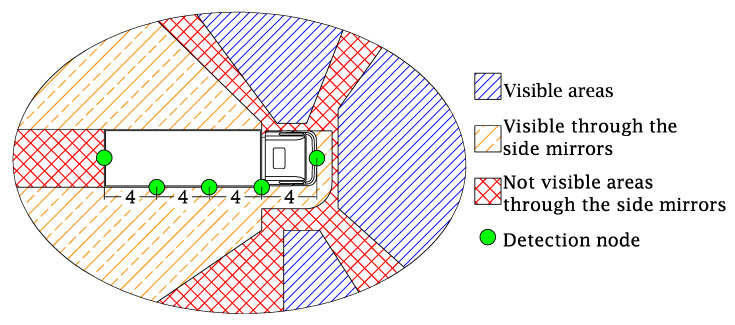
Truck with all detection nodes (green dots) mounted and all blind spots around a truck. Double blue lines are areas visible through the windshield, the orange solid and dashed lines are the areas visible through the side mirrors and the red crossed lines are not visible through the side mirrors, also known as the blind spot area.

**Figure 2 sensors-20-02727-f002:**
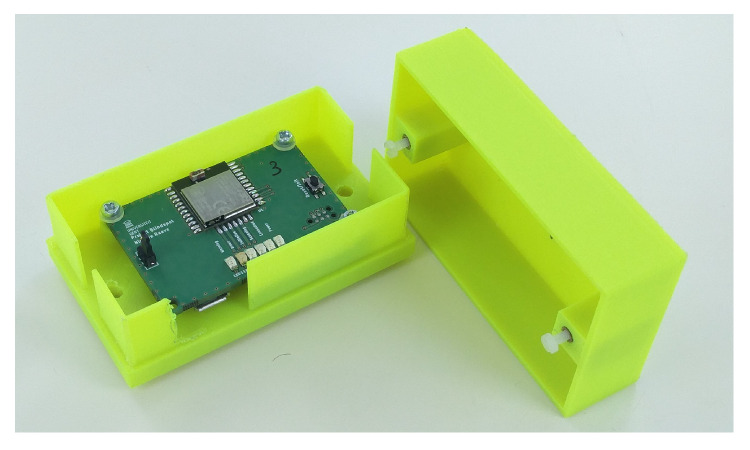
Designed detection node in its fluorescent housing.

**Figure 3 sensors-20-02727-f003:**
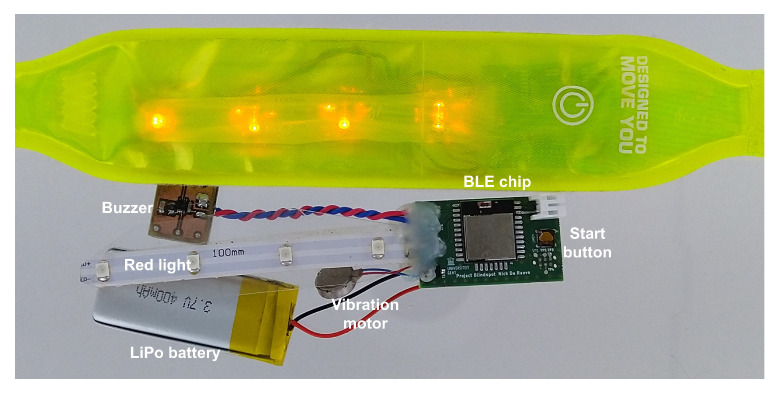
Designed wearable next to its housing.

**Figure 4 sensors-20-02727-f004:**
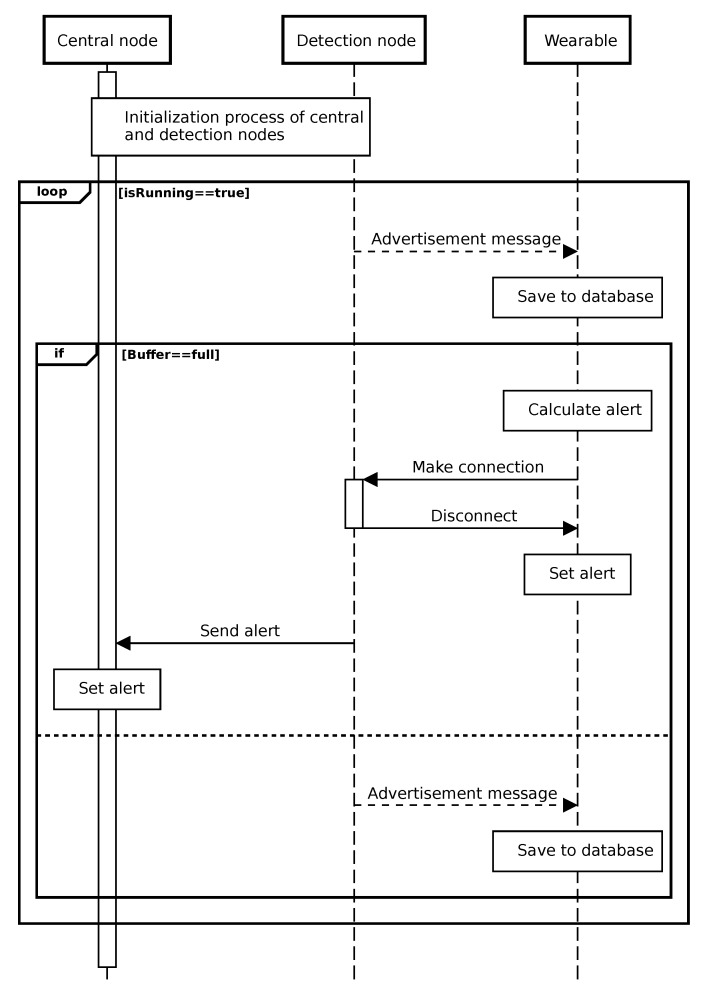
Sequence diagram of the designed system starting from the point when the wearable is activated.

**Figure 5 sensors-20-02727-f005:**
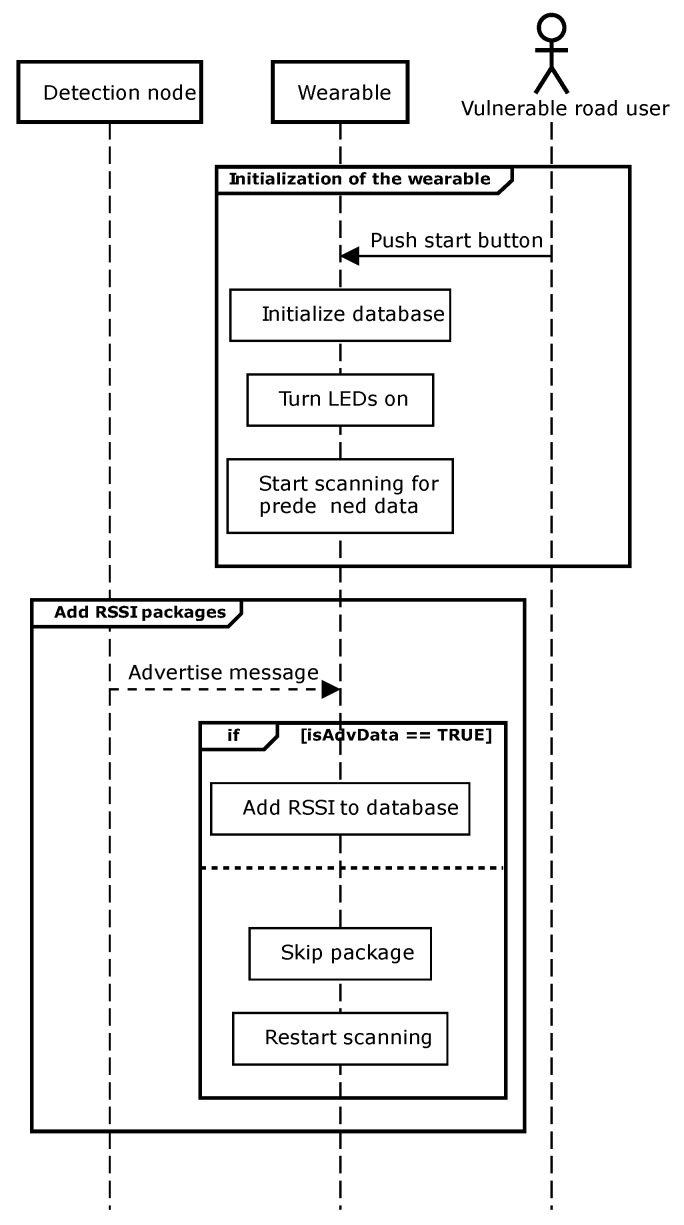
Sequence diagrams of the wearable with the initialization and add Received Signal Strength Indicator (RSSI) levels to the database software routine.

**Figure 6 sensors-20-02727-f006:**
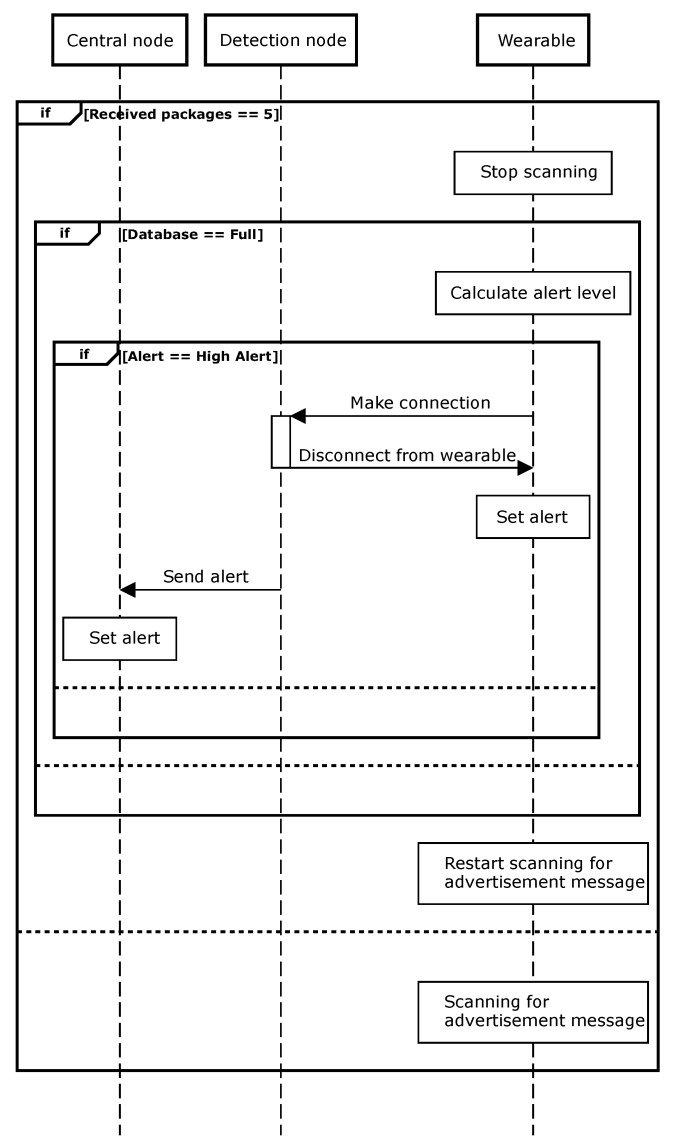
Sequence diagram of the wearable with the check database software routine.

**Figure 7 sensors-20-02727-f007:**
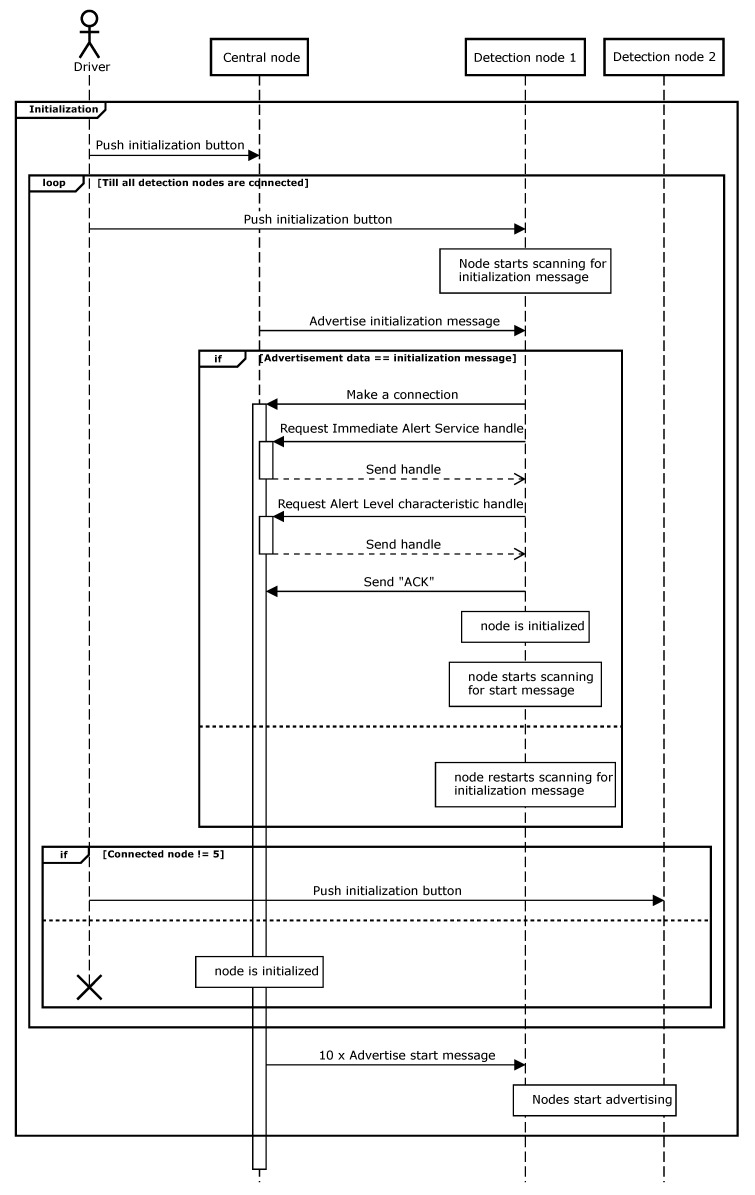
Sequence diagram of the initialization between detection node and central node.

**Figure 8 sensors-20-02727-f008:**
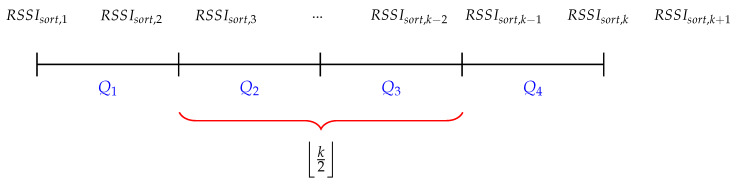
Graphical representation of the average filter with sliding window. k represents the buffer size, $Q_{1-4}$ represents the quartiles of the buffer.

**Figure 9 sensors-20-02727-f009:**
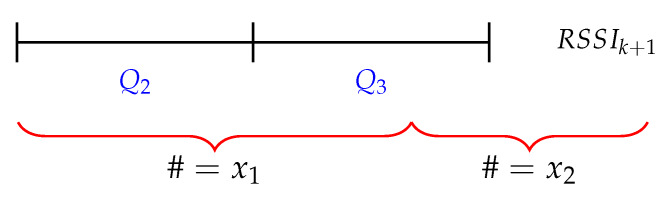
Graphical representation of the weight calculation.

**Figure 10 sensors-20-02727-f010:**
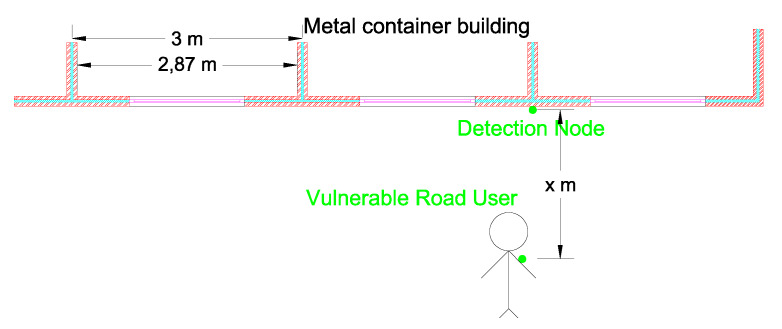
Top view of the static measurement setup against a metal container building. X represents the distance between the vulnerable road user with the wearable and the detection node mounted at a height of 1.2
m. X varies from 1 m to 8 m in steps of 1 m.

**Figure 11 sensors-20-02727-f011:**
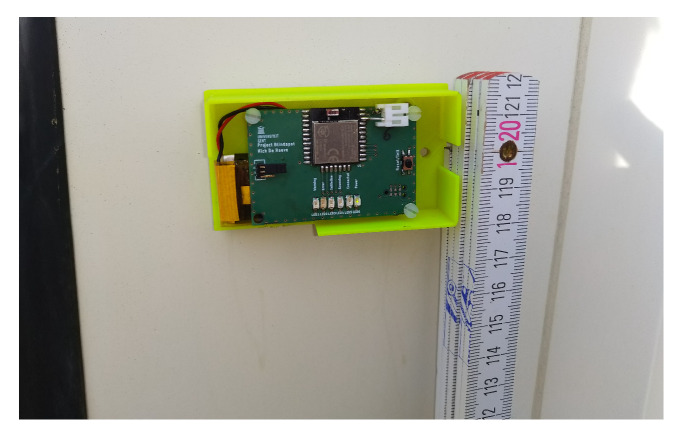
Detection node mounted against the metal container building at a height of 1.2
m.

**Figure 12 sensors-20-02727-f012:**
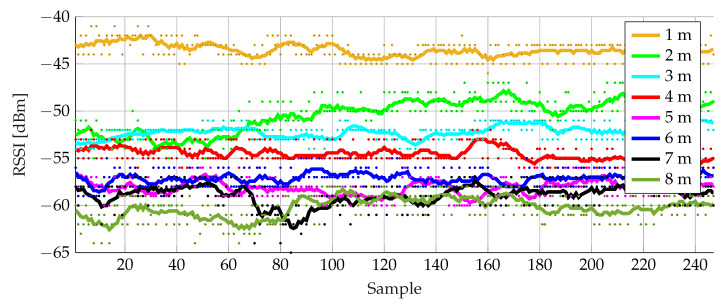
Averaged RSSI levels at distances of 1 m to 8 m in steps of 1 m.

**Figure 13 sensors-20-02727-f013:**
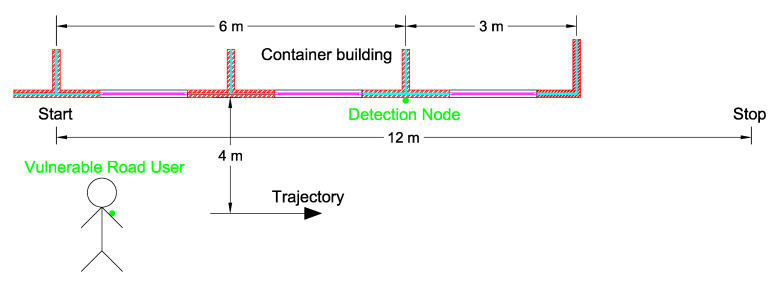
Top view of the dynamic measurement setup. The detection node is mounted at a height of 1.2
m. A test person wearing the wearable walks in front of the metal container building at a distance of 4 m. The test person follows a trajectory that starts at a distance of 6 m before and ends 6 m beyond the detection node.

**Figure 14 sensors-20-02727-f014:**
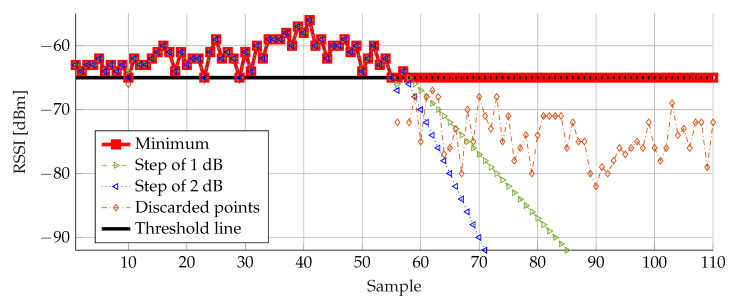
RSSI samples filtered by three different threshold filtering techniques.

**Figure 15 sensors-20-02727-f015:**
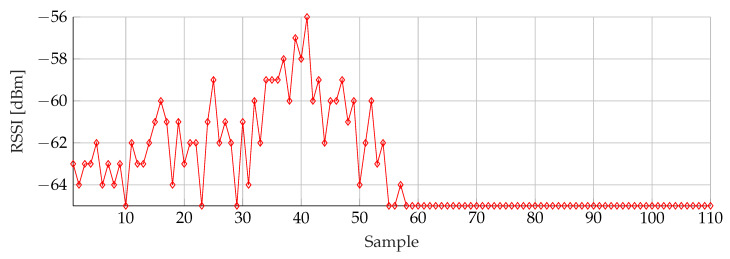
Minimum threshold filtered RSSI samples.

**Figure 16 sensors-20-02727-f016:**
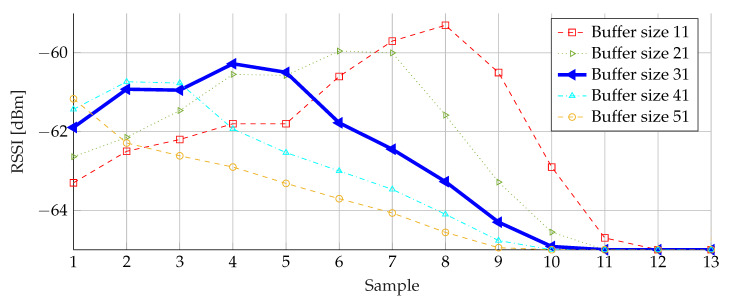
Weighted average filtering with different buffer sizes.

**Figure 17 sensors-20-02727-f017:**
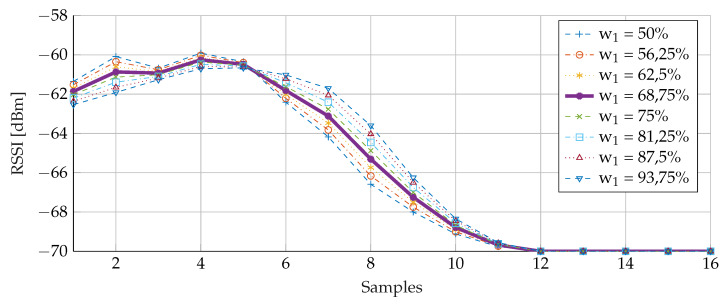
Weighted average filtering with different weights.

**Figure 18 sensors-20-02727-f018:**
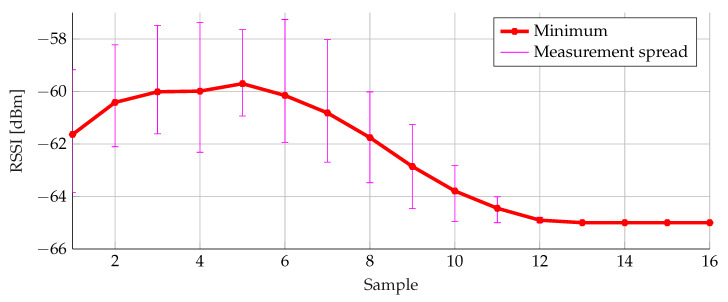
Threshold filtering with mean, minimum and maximum for three data sets.

**Figure 19 sensors-20-02727-f019:**
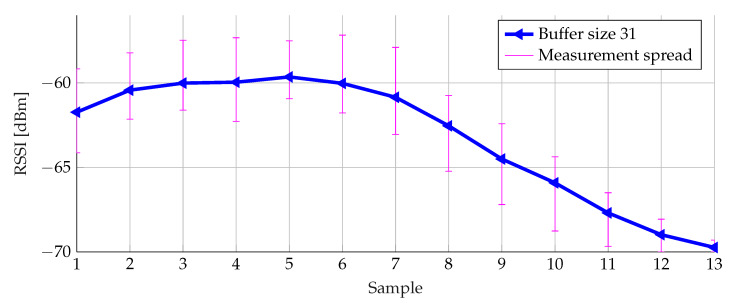
Weighted average filtering with different buffer sizes for three data sets.

**Figure 20 sensors-20-02727-f020:**
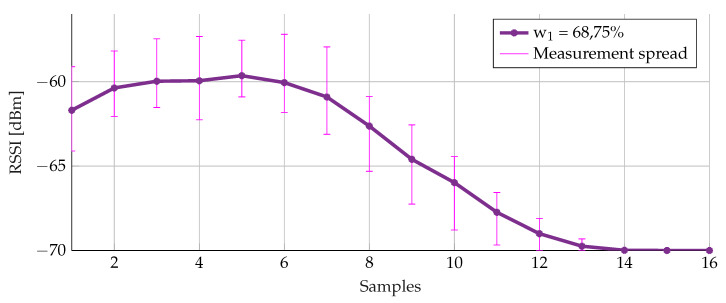
Weighted average filtering with different weights for three data sets.

**Figure 21 sensors-20-02727-f021:**
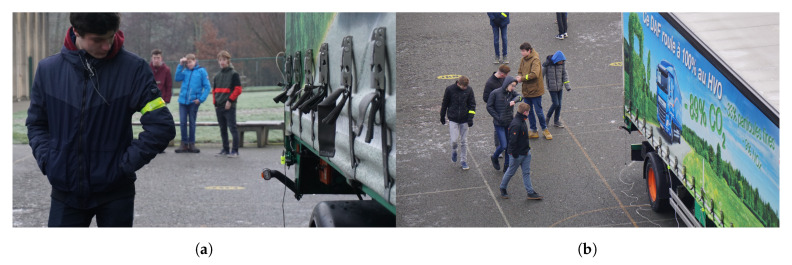
Pupils receiving alert during real-life measurements. (**a**) Pupil receiving second alert; (**b**) Group of ten receiving an alert.

**Figure 22 sensors-20-02727-f022:**
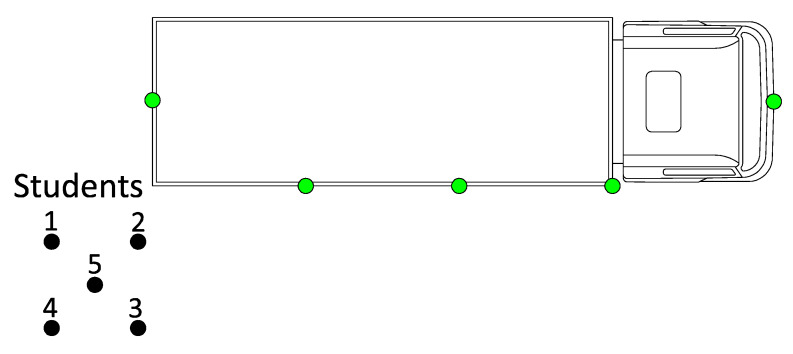
Top view of the real-life measurement test with big group of people. Five persons in a small group walked towards the truck starting at a distance of ±20 m behind the truck.

**Figure 23 sensors-20-02727-f023:**
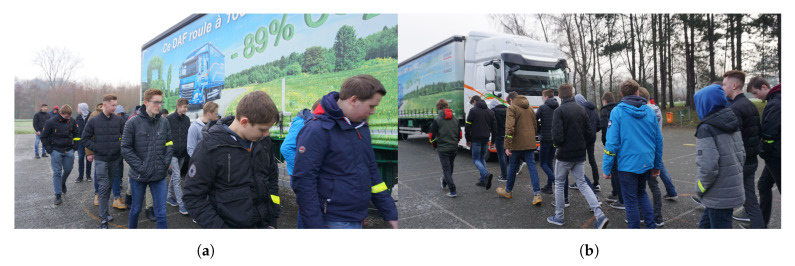
Group of twenty pupils receiving alert during measurement. (**a**) Start from behind truck; (**b**) Start from before truck.

**Figure 24 sensors-20-02727-f024:**
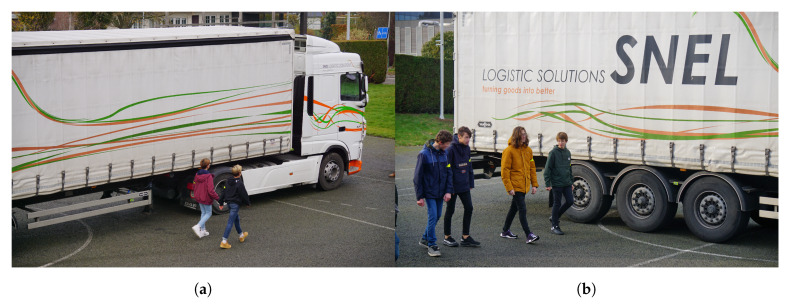
Pupils receiving alert during the verification measurement campaign. (**a**) Group of two received alerts; (**b**) Group of four received alerts.

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
