# Peer review of "A Bluetooth-Low-Energy-Based Detection and Warning System for Vulnerable Road Users in the Blind Spot of Vehicles"

_sensors, 2020, doi:10.3390/s20092727_

Round 1
Reviewer 1 Report
This paper presents the hardware implementation of a blind spot detection and warning system, relying on Received Signal Strength Indicator (RSSI) measurements and Bluetooth Low Energy (BLE) wireless communication. Also, the hardware implementation of different nodes and the design of a small sensor or wearable is presented.
1. The authors claim that "by studying multiple blind spot accidents, it is clear that, in Belgium, most vulnerable road users are on the right side and behind the truck," but they do not provide any reference to prove that.
2. It is not clear why authors choose the particular HW platform to implement the node. Was the decision based on the low-power operation, price, range, or something else? The authors should provide a short evaluation of the selected platform in terms of power/energy, range, and price and compare it with other competitive platforms.
3. Is it not clear why BLE 4.2 is chosen? The authors should justify this in terms of power/energy and range of BLE. Also, the authors do not provide any information on the BLE setup. I suggest the authors evaluate recent papers on BLE setup and energy performance. Please update your paper and comparison. For example:
a. Nikodem, M.; Bawiec, M. Experimental Evaluation of Advertisement-Based Bluetooth Low Energy Communication. Sensors 2019, 20, 107.
b. Bulić, P.; Kojek, G.; Biasizzo, A. Data Transmission Efficiency in Bluetooth Low Energy Versions. Sensors 2019, 19, 3746.
4. It is not clear why the window size of the moving average filter is 7.
5. The validation part (results) is incomplete. The authors should provide a comparison of the proposed solution with the state-of-the-art camera- and radar-based approaches to provide a clear picture of the key contributions of their results.
Author Response
2.11.0.0

Reviewer 2 Report
The paper provides a good warning mechanism for road users. It also analyses the impact of the RSSI and it presents a real prototipe.
In order to estimate the real performance, an evaluation of the effects of the buffer and filtering parameters is presented. Moreover, the paper identifies configuration challenges.
In my opinion, the authors presented a good work. However, I suggest some revisions to improve the paper.
I think that the real time measurement section should include more samples. It seems that it has only been perform one test.
Author Response
2.11.0.0

Reviewer 3 Report
The topic is about the blind spot road detection and warning system using BLE. The system is useful and interesting. However, there are some suggestions listed as follows.
1 The accuracy of detection is based on the RSSI levels, the design proposed a method to obtain the RSSI values by threshold-filter and slide-window filter. As the main contribution of this paper, this method is not quite innovative. There are many methods proposed in recent research on how to improve RSSI levels. I suggest the authors to compare the method with others to prove the effectiveness.
2 The design mentioned the nodes need to be connected before detection. The blind spots are limited, and the trucks are moving. Due to the speed and the area of the spot, the connection time is another significant key for the performance of the system. While the experiments on the latency are missing. I think it should be added and discussed.
Author Response
2.11.0.0

Round 2
Reviewer 1 Report
My comments have been addressed . In my opinion this version of the manuscript can be accepted for publication.
Reviewer 3 Report
All the issues are addressed.